# The Absence of Attentional Bias to Low-Calorie Food Stimuli in Restrictive Dieters: Differences in the Allocation of Attentional Resources to High-Calorie Foods

**DOI:** 10.3390/brainsci14060551

**Published:** 2024-05-29

**Authors:** Wu Jiang, Huan Zhang, Haoping Yang, Zonghao Zhang, Aijun Wang

**Affiliations:** 1Department of Psychology, Research Center for Psychology and Behavioral Sciences, Soochow University, Suzhou 215021, China; 2106403016@stu.suda.edu.cn (W.J.); 20215218040@stu.suda.edu.cn (H.Z.); 2School of Physical Education and Sports Science, Soochow University, Suzhou 215021, China; hpyang990306@stu.suda.edu.cn (H.Y.); zhangzonghao@suda.edu.cn (Z.Z.)

**Keywords:** restrictive dieters, attentional bias, attentional blink, RSVP

## Abstract

Restrictive dieters are those who expect to achieve body shape and weight control through dieting. However, they often have difficulty suppressing the desire to consume food when confronted with it. It has been shown that when high- and low-calorie foods are presented together, the attention of restrictive eaters is preferentially directed to high-calorie foods. However, whether attentional bias occurs when low-calorie foods are present alone and whether the allocation of attentional resources is consistent with that for high-calorie foods has yet to be explored. The present study focused on the effects of high-/low-calorie foods on attentional bias in restrictive dieters. Seventy-eight participants were recruited to participate in the experiment via the Dutch Eating Behavior Questionnaire (DEBQ) scale, which is administered in a rapid serial visual presentation (RSVP) task. The results revealed that failed restrictive dieters had the lowest percentage of correct answers at the lag2 level, indicating attentional bias. Failed restrictive dieters allocated more attentional resources to high-calorie foods than to low-calorie foods. Restrictive dieters showed no attentional bias when low-calorie foods were presented alone. The results suggest that low-calorie foods do not elicit an attentional bias in restrictive dieters and that the allocation of attentional resources is not consistent when compared to that for high-calorie foods.

## 1. Introduction

Restrained eaters are dieters who have long-term intentions to lose weight [1]. In reality, individuals prioritize attention to food-related stimuli, a phenomenon known as food attention bias [2]. It has been shown that restrictive dieters are more influenced by food-related types of stimuli and have a greater attentional bias for food stimuli than nonrestrictive dieters who have no intention of losing weight [3,4]. Restrictive eaters hope to achieve weight and body image control through dieting [5] but are usually unsuccessful; they initially have a period of inhibition to control their eating, and when they can no longer inhibit their eating, they binge [6]. Restrictive eaters can quickly turn their attention to high-calorie foods [7] and have difficulties with attentional disengagement [8]; therefore, they may have difficulty maintaining the strategy of inhibiting eating over the long term, leading to uninhibited eating and, consequently, eating-related behaviors.

Previous studies on food attention bias in restrictive dieters have explored it in the spatial/temporal dimension using different paradigms and methods [9,10,11]. Restrictive dieters’ attentional bias toward food is influenced by the caloric content of the food, and high-calorie foods, in particular, can capture their attention more quickly [9,12,13]. This may explain why restrictive dieters sometimes cannot control the desire to eat after a period of strict dieting, thus encouraging binge eating behaviors. Low-calorie foods do not have a large effect on the attentional bias of restrictive dieters, possibly because low-calorie foods do not have as large an effect on body weight as high-calorie foods, and restrictive dieters are not as sensitive to low-calorie foods as they are to high-calorie foods [14]. In flanker tasks, restrictive dieters are also attracted to high-calorie foods when low-calorie foods are used as targets and high-calorie foods are used as flankers [12].

The above studies explored the food attention bias of restrictive dieters in the spatial dimension. In the temporal dimension, individuals’ attention is often measured using the Rapid Serial Visual Presentation (RSVP) task [15,16,17]. The basic component of the RSVP paradigm is the presentation of a rapidly flashing stream of 10–19 stimuli (e.g., 118 ms/stimulus) on a screen [3,9], with no interstimulus interval between stimuli, in which 1 or 2 stimuli may be the target stimulus, and the lag (time) between the 2 target stimuli can be controlled. Numerous studies have demonstrated that when the 2nd target stimulus (Target 2, T2) appears within 500 ms after the appearance of the first target stimulus (Target 1, T1), individuals often do not observe T2, a phenomenon known as attentional blink (AB). When T2 appears at a longer lag (>500 ms), individuals have less difficulty recognizing T2 [18]. The phenomenon of AB reflects the limitations of individual attention in processing stimuli in the temporal dimension.

One study revealed that when food was presented as T1, the attentional blink effect was enhanced in restrictive dieters, leading to a reduction in the recognition of nonfood stimuli as T2. In addition, when food stimuli were presented as T2, recognition of previous nonfood T1 targets was impaired [9]. In another study, a reduction in the recognition of neutral targets was found when food, as a distractor, was presented before a nonfood target, which was not found in unrestrictive dieters. In addition, it was similarly found that the recognition of T1 was reduced when food served as T2 [3]. These findings suggest the phenomenon of prioritization of food targets in restrictive dieters and reverse interference with previous targets.

Higher-calorie foods have greater hedonic attributes [19] and are more likely to elicit an attentional bias in restrictive eaters [20]; therefore, the attentional bias of restrictive eaters toward higher-calorie foods has been primarily explored in previous research on the RSVP paradigm. There are differences in exploring the effects of high-/low-calorie foods on restrictive dieters’ attentional bias in the spatial dimension paradigm. However, it is not known whether there are differences in the temporal dimension between foods of different levels of caloric content due to restrictive dieters’ attentional bias, especially when high-/low-calorie foods are presented separately. Therefore, in the present study, we investigated the effects of high-/low-calorie foods on the attentional bias of restrictive dieters in the time dimension. Our goal was to understand whether low-calorie foods create an attentional bias in restrictive eaters and the extent to which the caloric intake level affects attentional bias in restrictive eaters. The experiment was conducted in an RSVP paradigm, with a high- or low-calorie food presented randomly as T1 and neutral stimuli presented as T2. The hypotheses were that participants would have the largest AB effect at the lag2 level in the high-/low-calorie food stimulus condition and that low-calorie foods would not induce an attentional bias in restrictive dieters.

## 2. Methods

### 2.1. Participants and Measurement Tools

This study was adopted as a 3 × 3 experimental design, and the data were processed through repeated measures ANOVA. The sample size was estimated using the computational software G-Power 3.1 (estimated effect size f = 0.2, α = 0.05, power = 0.9), which predicted a minimum total sample size of 63 participants, and each group recruits a minimum of 21 participants. In the actual study, we actually recruited 85 participants, taking into account issues such as sample attrition. Inclusion criteria were no problems with vision, color blindness, or color deficiency, no history of mental illness, and a body mass index (BMI) between 18.5 and 24.9 [21]. Participants with myopia wore eyes during the course of the experiment. A total of 7 participants were excluded due to very low correctness rates (i.e., the identification rate was less than 2 standard deviations below the sample mean), and the data from 78 participants were ultimately included in the final sample (56 females and 22 males with an age range of 18~25 and a mean age of 19.7 years). There were 46 liberal arts majors and 32 science majors. Through the questionnaire, those who failed restrictive diets and those who succeeded restrictive diets were in a dieting state, and those who were not in a non-restrictive diet were in a non-dieting state and had no diet-related problems (see Table 1). The participants were all undergraduate students who were normal-weight and healthy. We strictly adhered to the basic principles of the Declaration of Helsinki, and participants volunteered to participate in the experiment and signed an informed consent form. Each participant was paid 30 RMB at the end of the experiment, and they were unaware of the purpose of the experiment during the course of the experiment.

The Dutch Eating Behavior Questionnaire (DEBQ) [22], a 33-item scale consisting of a restrictive eating score and an inhibitory eating score, was used as the measurement instrument. Van Strien et al. stated that the mean score of the questionnaire for the “Restrictive Eating” section was the “Restrictive Score,” and the mean score of the questionnaire for the “Emotional Eating” and “External Eating” sections was the “Suppressive Score.” The average of the scores for the “external eating” questionnaire was the “inhibitory score.” Participants who met the mean scores of the “Restriction Score” and “Inhibition Score” ≥ 3 were failed restrictive eaters; those who met the “Restriction Score” ≥ 3 and the “Inhibition Score” ≥ 3 and the “Inhibition Score” ≥ 3 were failed restrictive eaters. “Inhibitory Score” < 3 were successful restrictive dieters; those with “Restrictive Score” < 3 were non-restrictive dieters [23].

A person with a mean restriction score < 3 is considered a nonrestrictive eater (UN_RE), a person with a mean restriction score ≥ 3 and a mean inhibition score < 3 is considered a successful restrictive eater (S_RE), and a person with both restriction and inhibition scores ≥ 3 is considered a failed restrictive eater (F_RE) [24]. A five-point scale was used, where 1 means “never” and 5 means “always”. The internal consistency coefficients (Alpha) for the restriction and disinhibition scores were 0.83 and 0.89, respectively.

There were 26 participants in each group. There was no significant difference in age among the three groups of participants: *F*(2, 3.16) = 1.22, *p* > 0.05, *η_P_*^2^ = 0.03. There was a significant difference in restriction scores among the three groups of participants: *F*(2, 13.03) = 62.98, *p* < 0.001, *η_P_*^2^ = 0.62; failed restrictive dieters (3.43) had higher restriction scores than nonrestrictive dieters (2.25), *t*(52) = 7.41, *Cohen’s d* = 2.39, *p* < 0.001, 95% CI = [0.92, 1.42]; successful restrictive dieters (3.53) had higher restriction scores than nonrestrictive dieters (2.25), *t*(52) = 10.15, *Cohen’s d* = 2.61, *p* < 0.001, 95% CI = [1.02, 1.52]; and there was no significant difference in restriction scores between successful restrictive eaters (3.53) and failed restrictive eaters (3.43), *p* > 0.05. There was a significant difference in the dew-suppression scores among the three groups of participants, *F*(2, 7.26) = 38.54, *p* < 0.001, *η_P_*^2^ = 0.50; failed restrictive eaters (3.43) had greater disinhibition scores than did unrestrictive eaters (2.61), *t*(52) = 5.71, *Cohen’s d* = 1.78, *p* < 0.001, 95% CI = [0.58, 1.06], failed restrictive diets (3.43) had greater disinhibition scores than did successful restrictive dieters (2.44), *t*(52) = −0.59, *Cohen’s d* = 2.46, *p* < 0.001, 95% CI = [1.22, 0.74]; there was no significant difference (*p* > 0.05) between successful restrictive dieters (2.44) and nonrestrictive dieters (2.61) in terms of dew suppression scores.

### 2.2. Apparatus and Materials

The programming software Python 3.12.3 3 was used to program the experiment; the stimuli were displayed on a 14-inch display with a screen resolution of 1920 × 1080 pixels; PsychoPy3.2.4 (Psychology Experiment Software) was used for the presentation of the experiment and the collection of data; and the experiment was run in a Microsoft Windows 11 system environment.

The stimulus material was obtained from the “Food-pics” database [25], which is a collection of images containing both food and nonfood pictures. In the study, T1 consisted of two images of food items from each of the two levels of caloric content (high-calorie and low-calorie), with the food calorie information being as follows: burger (258 calories/100 g), roast chicken (200 calories/100 g), pear (51 calories/100 g), and grapes (69 calories/100 g). T2 consisted of nonfood images (e.g., cat, potted plant, chair) with 98 neutral images (consisting of images from the household/kitchen tools category) that appeared as distractor stimuli, for a total of 102 stimulus pictures.

Prior to the experiment, 142 normal-weight college students with no diet-related problems were recruited to rate the materials used in the experiment for emotional arousal, emotional potency, and favoritism. High-calorie and low-calorie foods were grouped together, and a one-way ANOVA revealed no significant between-group effects; there were no significant differences in emotional arousal (*p* = 0.785), emotional potency (*p* = 0.626), or favoritism (*p* = 0.603) between the 2 high-calorie food pictures and the 2 low-calorie food pictures (see Table 2).

### 2.3. Design and Procedure

The present study utilized a two-factor experimental design, which allowed for the highlighting of group differences in participants, a more in-depth understanding of the cognitive mechanisms of those on restrictive diets, and a reduction in potential confounders in the experiment, making the findings more relevant.

The experiment used a 3 (lag: lag2/lag4/lag7) × 3 (restriction type: successful restrictive dieter/failed restrictive dieter/nonrestrictive dieter) mixed design. The within-participants factor was lagged, the between-participants factor was group, and the dependent variables were T1 correctness and T2|T1 correctness (correctness of T2 given the judgment that T1 is correct). The experiments were conducted in a completely randomized design, and the experiments were divided into 6 groups, with a total of 48 trials for each combination of lag and heat conditions in 1 group of experiments, for a total of 288 trials in the 6 groups. Participants were given a rest period after each group of experiments.

At the beginning of the experiment, a 1000 ms cross-gaze point was presented at the center of the screen (see Figure 1), and the participants were asked to intently stare at the center of the screen. Then, a stimulus stream consisting of 12~19 pictures was presented rapidly at a rate of 118 ms each. T1 was a stimulus stream consisting of 2 pictures of high-calorie food and 2 pictures of low-calorie food, for a total of 4 images, which appeared randomly at either position 6 or position 8 in the stimulus stream. T2 was a set of nonfood neutral picture stimuli presented at 236 ms, 472 ms, and 826 ms after the presentation of T1, i.e., lag2, lag4, and lag7, respectively. After a set of stimulus streams had been presented, participants were asked to answer two questions (Q1: the direction of rotation of the first yellow-framed target and Q2: the direction of rotation of the second yellow-framed picture), and participants judged left and right by pressing the left and right arrow keys of the keyboard. Prior to the formal experiment, we had participants participate in 13 practice trials, which had the same content as the formal experiment.

### 2.4. Data Analyze

This experiment uses WPS Office version 2024 software to organize and analyze the experimental data and will use IBM SPSS Statistics 26 software to statistically analyze and test the valid data, with a significance level of α = 0.05. The data analysis mainly contains two main indicators: the T1 correct rate (the probability of correctly judging the first target) and the T2|T1 correct rate (the probability of correctly judging the second target if the first target is correctly judged). Main and interaction effects were tested by repeated measures ANOVA, and simple effects analyses were used to compare differences between factors in specific conditions, with Bonferroni corrections when needed. In the experiment, the independent variables were lag (lag2/lag4/lag7) and the calorie content of the food (high/low), the dependent variable was the percentage of participants T2|T1 who were correct, the lag condition was a within-groups factor, and the three groups of participants were between-groups factors.

## 3. Results

### 3.1. Analysis of the T2|T1 Correctness in the Three Groups of Participants in the High-Calorie Condition

The results of the ANOVA using 3 × 3 repeated measures revealed that the lag condition main effect was significant in the high-heat condition (see Table 3), *F*(2, 74) = 117.28, *p* < 0.001, *η_P_*^2^ = 0.76, suggesting that participants’ correctness rates were different across lag conditions, i.e., the longer the lag, the higher the correctness rate of the participants, which is typical of the AB effect. Further pairwise comparisons revealed that the correct rate at the lag2 level (72.3%) was significantly lower than that at the lag4 level (86.6%), *t*(52) = −12.12, *Cohen’s d* = −1.16, *p_bonf_* < 0.001, 95% CI = [−0.16, 0.11]; the correct rate at the lag2 level (72.3%) was significantly lower than that at the lag7 level (90%), *t*(52) = −15.11, *Cohen’s d* = −1.72, *p_bonf_* < 0.001, 95% CI = [−0.21, 0.16]; and the correct rate at the lag4 level (86.5%) was significantly lower than that at the lag7 level (90%), *t*(52) = −4.28, *Cohen’s d* = −0.43, *p_bonf_* < 0.001, 95% CI = [−0.06, 0.02].

The main effect between groups was significant (see Table 3), *F*(2, 0.11) = 4.72, *p* = 0.012, *η_P_*^2^ = 0.11, suggesting differences in correctness rates between groups. Further pairwise comparisons revealed that nonrestrictive dieters (87.2%) had significantly greater correct response rates than did failed restrictive dieters (79.7%, *t*(52) = −4.46, *Cohen’s d* = −0.65, *p_bonf_* < 0.001, 95% CI = [0.04, 0.10], successful restrictive dieters (82.8%) had significantly greater correct responses than did failed restrictive dieters (79.6%), *p_bonf_* = 0.068, and nonrestrictive dieters (87.2%) had greater accuracy rates than did successful restrictive dieters (82.8%), *t*(52) = −2.75, *Cohen’s d* = −0.43, *p_bonf_* = 0.007, 95% CI = [−0.07, 0.01].

The lag × group interaction effect was significant (see Table 3), *F*(2, 75) = 4.14, *p* = 0.020, *η_P_*^2^ = 0.1, suggesting that there were differences in the percentage of correctness of participants in different groups at each level of the lagged condition (see Figure 2A). Further simple effects analysis revealed that at the lag2 level, failed restrictive dieters (67.9%) had significantly lower correct rates than nonrestrictive dieters (76.7%), *t*(52) = −2.64, *p_bonf_* = 0.044, *Cohen’s d* = −0.73, 95% CI = [−0.18, 0.02]; at the lag4 level, those failed restrictive dieters (81.7%) had significantly lower correct rates than unrestrictive dieters (91.8%), *t*(52) = −3.5, *p*_bonf_ = 0.003, *Cohen’s d* = −0.97, 95% CI [−0.18, 0.03]; at and at the lag7 level, there was no significant difference in correctness among the three groups, *p_bonf_* > 0.05.

### 3.2. Analysis of the T2|T1 Correctness in the Three Groups of Participants in the Low-Calorie Condition

Using a 3 × 3 repeated-measures ANOVA, a significant main effect of lagged condition was found (see Table 4), *F*(2, 74) = 101.16, *p* < 0.001, *η_P_*^2^ = 0.73, and further pairwise comparisons revealed that lag2 (67.2%) was significantly less correct than lag4 (78.7%), *t*(52) = −10.26, *p_bonf_* < 0.001, *Cohen’s d* = −1.16, 95% CI [−0.14, −0.09]); lag2 (67.2) was significantly lower than lag7 (83%), *t*(52) = −14.41, *p_bonf_* < 0.001, *Cohen’s d* = −1.63, 95% CI [−0.19, −0.13]); and lag4 (78.7%) had a significantly lower correct rate than lag7 (83%), *t*(52) = −4.59, *p_bonf_* < 0.001, *Cohen’s d* = −0.52, 95% CI [−0.06, −0.02]). The main effect between groups was not significant, *F*(2, 0.06) = 2.23, *p* = 0.114, and the lagged × group interaction effect was not significant, *F*(2, 75) = 1.21, *p* = 0.304 (see Figure 2B).

## 4. Discussion

In this study, participants were categorized using the DEBQ [22] and divided into three groups (successful restrictive eaters, failed restrictive eaters, and nonrestrictive eaters) based on their restriction and disinhibition scores on the scale, and the attentional bias of the three groups of participants was explored in an RSVP paradigm. The main objective was to explore the effects of and differences in foods with different levels of caloric intake on attentional bias in restrictive dieters. The results of the experiment revealed the following: First, the AB disengagement phenomenon exists, as evidenced by the fact that participants had the lowest rate of correctness at the lag2 level, and this correctness level increased as the lag time increased. Second, there was a difference in participants’ recognition of high- and low-calorie foods, with the correct rate for high-calorie foods being significantly greater (88.9%) than that for low-calorie foods (83%), suggesting that high-calorie foods can be processed and recognized more quickly in working memory [9,12,13]. Third, there was a difference in the rate of correctness among the three groups of participants. Specifically, nonrestrictive dieters had the highest rate of correctness, and failed restrictive dieters had the lowest rate of correctness. Fourth, failed restrictive dieters had more attentional resources for recognizing high-calorie foods, resulting in lower rates of correctness at the lag4 level.

Participants were least correct at the lag2 level in the high- and low-calorie food conditions, and correctness increased with increasing lag time (beyond the window of the AB), which is typical of an AB effect and consistent with previous findings [3]. This finding corroborates the two-stage model, where T1 enters the second stage, the consolidation stage, when all items have been initially processed. When T2 appears in the second stage of T1, T2 is disturbed and unable to enter the consolidation stage, which results in its relatively unstable representation. The subsequent emergence of nontarget stimuli by T2 subsequently overrides T2’s unstable representation, preventing T2 from being correctly recognized and reported, leading to the AB phenomenon [18].

In the high-calorie condition, the attention of the failure group was attracted to the high-calorie food stimuli and further processed them [26], allocating more attentional resources to the high-calorie food stimuli [19]; this can interfere with the identification of subsequent targets, thereby increasing the AB [27], which is in line with the results of a previous study [3]. Using a two-stage model, this can be explained by the fact that when failed restrictive eaters occupied more attentional resources in the first stage of recognizing high-calorie foods, this led to a delay in the second stage of consolidation, i.e., T2 in the T1 versus lag2 condition was not efficiently consolidated. Failed restrictive dieters are more likely to focus on high-calorie foods. This can be explained by the strength of the desire to eat. The desire to eat and restrictive eating are negatively correlated. A stronger desire to eat means that restriction is more likely to fail, which enhances attentional bias towards food, meaning that the desire to eat and attentional bias are positively correlated [28]. Further, a higher desire to eat breaks down inhibitions in those who have failed a restrictive diet and increases attentional bias. Another explanation is that the success and failure groups differed in their cognitive strategies for dealing with food stimuli. The absence of an attentional bias toward high-calorie foods in the successful restrictive dieter group may be because they attempted to inhibit food cravings through a cognitive strategy in which the individual desires to approach and consume high-calorie foods but is also concerned about the negative consequences of consuming high-calorie foods [29], an attentional pattern that may help them to maintain an inhibited state of eating. Failed restrictive eaters, on the other hand, may not be able to suppress the desire to eat, maintaining a persistent attention to high-calorie foods [30].

The time interval between lag4 and T1 was 472 ms, which is a marginal time for AB production, or, in other words, a time during which AB production is more susceptible to group influence. In the present study, regarding the performance of the failed restrictive dieters on lag4, although there was an improvement relative to that on lag2, indicating that there was still attentional maintenance, i.e., the attentional bottleneck was not in a state of complete idleness [31]. This finding suggests that the failed restrictive dieter group allocated more attentional resources to and had more difficulty shifting attention away from the high-calorie food stimuli [8]. This sustained attention to high-calorie foods prevents subsequent stimuli from entering working memory.

In the low-calorie condition, there was no significant difference in the percentage of correctness of any of the three groups of participants; i.e., the effect of low-calorie food on the three groups of participants was consistent, and low-calorie foods do not have the hedonic property of enhancing working memory encoding compared to high-calorie foods [32]. Restrictive diets primarily restrict high-calorie foods, and when individuals consume low-calorie foods frequently over a period of time due to their lack of hedonicity, seeing low-calorie foods again may lead to avoidance [31,33], prohibiting the creation of attentional bias [34].

Differences in the recognition of high- and low-calorie food stimuli exhibited by restrictive and unrestrictive dieters in the present study suggest that the caloric content of the food influences the allocation of attentional resources in restrictive dieters, with high-calorie foods being a more salient cue, especially for failed restrictive dieters [20]. Restrictive eaters are more sensitive to cues they perceive as conflicting with their dieting goals (e.g., high-calorie foods), and this hypersensitivity leads to greater focus, making it difficult to disengage from these stimuli.

The present study discussed the effect of the caloric content of the food on the attentional bias of restrictive eaters in the RSVP paradigm, and the experimental results demonstrated that low-calorie foods do not elicit an attentional bias in restrictive eaters and have a limited effect on the attentional bias of restrictive eaters compared to high-calorie foods. This also hints at the phenomenon that when restrictive eaters are in a period of dieting, they forcibly suppress their desire for high-calorie foods. According to Blechert, this desire to eat becomes stronger the more it is suppressed, resulting in a state of hedonic deprivation [35]. Hedonic deprivation is exposed when stimuli related to high-calorie foods are present in the external environment, which in turn affects individuals’ behavior. The fact that we live in an environment with easy access to food makes it potentially more difficult for restrictive dieters to achieve their dieting goals. They must curb their desire for high-calorie foods, but this desire can be ameliorated through the use of interventional training and orthomolecular therapies to reduce the attentional bias toward high-calorie foods in restrictive dieters [10,36,37,38,39]. Establishing a healthy mindset and good beliefs can help restrictive dieters achieve their dieting goals while maintaining a healthy body.

### Possible Limitations of This Study

The content of our study focuses mainly on the caloric content of the food and may overlook other important factors that may affect attentional bias, such as the palatability of and familiarity with the food and the failure to consider the relevance of the hedonic properties of food and individual olfactory sensitivity. The results of the study may be affected by the relatively small sample size. The participants in the sample were mainly from specific demographic and educational backgrounds, and these factors may affect the breadth of the experimental results. This study may not have fully controlled for a number of external variables that may affect attention, such as stress levels and emotional states. These factors may have influenced participants’ responses to food-related stimuli.

## 5. Conclusions

The present study explored the attentional bias of restrictive eaters towards high/low calorie foods in a rapid serial visual presentation (RSVP) paradigm with high/low calorie foods as T1 and neutral stimuli as T2. The group was divided into three groups: successful restrictive dieters, failed restrictive dieters, and nonrestrictive eaters. It was found that successful restrictive dieters and failed restrictive dieters had no attentional bias towards low-calorie foods; while failed restrictive dieters had an attentional bias towards high-calorie foods. This suggests that successful and failed restrictive eaters have different perceptions as well as coping strategies when confronted with high-calorie foods, which results in failed restrictive eaters having more difficulty suppressing the desire to eat high-calorie foods.

## Figures and Tables

**Figure 1 brainsci-14-00551-f001:**
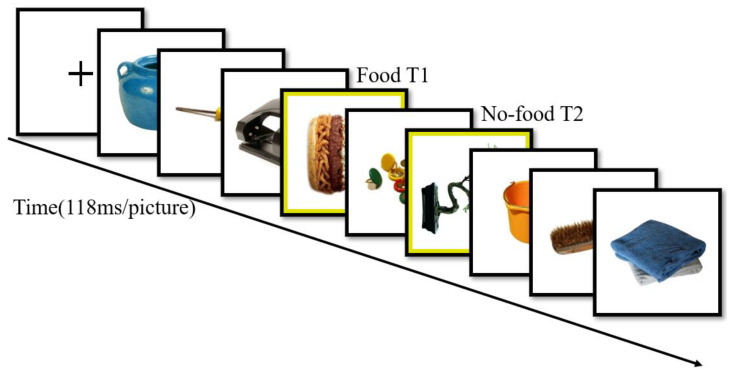
The figure shows one trial of an experiment in which pictures were presented uninterruptedly at 118 ms/sheet. T1 appeared at the 6th or 8th position in the visual stimulus stream (after 5 or 7 interferences). T2 appeared at the 2nd, 4th, or 7th position after T1 (with 1, 3, or 6 intervening interferences). T2 appeared at the 2nd, 4th, or 7th position after T2 (with 1, 3, or 6 intervening interferences). The participants were asked to recognize the T1 and T2 targets.

**Figure 2 brainsci-14-00551-f002:**
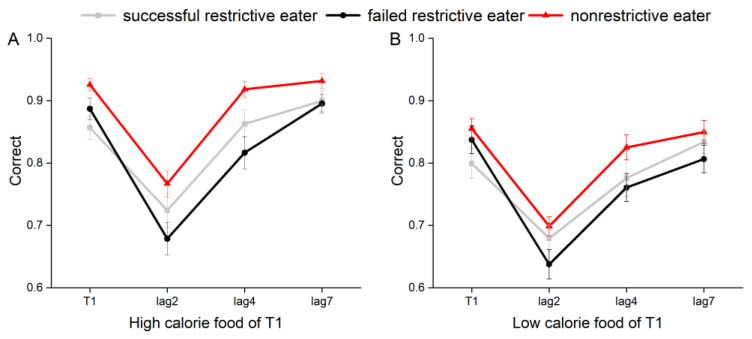
(**A**) shows participants’ T2|T1 (correctness of T2 given the judgment that T1 is correct, T2 occurs at lag2/lag4/lag7) correctness in the high-heat condition, and (**B**) shows participants’ T2|T1 correctness in the low heat condition; all three groups of participants had the lowest correctness in both heat conditions at the lag2 level, and the correctness increased as the lag increased. Error bars are the error values of the means.

**Table 1 brainsci-14-00551-t001:** Demographic information of the participants (M ± SD).

Demographic Information	S_RE (*n* = 26)	F_RE (*n* = 26)	UN_RE (*n* = 26)	*F*	*p*
Age	19.31 ± 1.19	19.85 ± 1.61	19.96 ± 1.92	1.22	0.300
Restricted scores	3.53 ± 0.36	3.43 ± 0.37	2.25 ± 0.59	62.98	0.001 **
De-emphasis scores	2.44 ± 0.36	3.43 ± 0.44	2.61 ± 0.48	38.54	0.001 **
year of study	1.92 ± 0.84	2.27 ± 1.11	2.19 ± 1.02	0.85	0.428

Note: S_RE mean successful restrictive eater; F_RE mean failed restrictive eater; UN_RE mean nonrestrictive eater. ** = *p* < 0.001.

**Table 2 brainsci-14-00551-t002:** Validity, arousal, and favorability ratings of experimental materials.

Variant	Food Category	M ± SD	*F*	*p*
potency	high-calorie food	5.97 ± 2.01	0.23	0.626
	low-calorie food	6.08 ± 1.87		
arousal	high-calorie food	5.89 ± 2.09	0.07	0.785
	low-calorie food	5.96 ± 2.02		
preference	high-calorie food	6.09 ± 1.95	0.27	0.603
	low-calorie food	6.20 ± 1.80		

**Table 3 brainsci-14-00551-t003:** Comparison of correct rates (%) for main, between-group, and interaction effects in high-calorie conditions.

Correct Rate (%)	*t*(52)	*Cohen’s d*	*p_bonf_*	95% CI
Lag2 (72.3) < Lag4 (86.6)	−12.12	−1.16	0.001	[−0.16, 0.11]
Lag2 (72.3) < Lag7 (90)	−15.11	−1.72	0.001	[−0.21, 0.16]
Lag4 (86.6) < Lag7 (90)	−4.28	−0.43	0.001	[−0.06, 0.02]
UN_RE (87.2) > F_RE (79.7)	−4.46	−0.65	0.001	[0.04, 0.10]
UN_RE (87.2) > S_RE (82.8)	−2.75	−0.43	0.007	[−0.07, 0.01]
S_RE (82.8) > F_RE (79.7)			0.068	
Lag2: F_RE (67.9) < UN_RE (76.7)	−2.64	−0.73	0.044	[−0.18, 0.02]
Lag4: F_RE (81. 7) < UN_RE (91.8)	−3.5	−0.97	0.003	[−0.18, 0.03]
Lag7:			>0.05	

**Table 4 brainsci-14-00551-t004:** Comparison of lagged main effect correctness (%) under low heat conditions.

Correct Rate (%)	*t*(52)	*Cohen’s d*	*p_bonf_*	95% CI
Lag2 (67.2) < Lag4 (78.7)	−10.26	−1.16	0.001	[−0.14, −0.09]
Lag2 (67.2) < Lag7 (83)	−14.41	−1.63	0.001	[−0.19, −0.13]
Lag4 (78.7) < Lag7 (83)	−4.59	−0.52	0.001	[−0.06, −0.02]

## Data Availability

The data that support the findings of this study are available from the corresponding authors (A.W.) upon reasonable request. The data are not publicly available due to privacy.

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
