# Peer review of "The Absence of Attentional Bias to Low-Calorie Food Stimuli in Restrictive Dieters: Differences in the Allocation of Attentional Resources to High-Calorie Foods"

_brainsci, 2024, doi:10.3390/brainsci14060551_

Round 1

Reviewer 1 Report

Comments and Suggestions for Authors

The manuscript "The absence of attentional bias to low-calorie food stimuli in restrictive dieters: Differences in the allocation of attentional resources to high-calorie foods" could be an interesting experimental study, if study design would be appropriate. However, I have many methodological concerns about the manuscript.

1. The authors did not provide for what statistical test was used in GPower software. If it was ANOVA, it should be clear what type (one-way, two-way, repeated measures) and how many groups and repetitions were expected. The sample size for each group should be calculated, as well as for the total sample.

2. How "normal or corrected-to-normal vision, no color deficiency or color blindness, and no history of psychiatric illness" was detected?

3. How many participants were included in each group: S_RE, F_RE, and UN_RE?

4. How "normal weight and health" were assessed?

5. More information is required about participants: (a) year of study; (b) faculty and study major; (c) current dieting (e.g., no diet, vegetarian, keto, etc.); and (d) eating disorder (e.g., bulimia, anorexia, binge eating, etc.). These information is necessary to replicate the study and justify the sample as appropriate for the study purpose.

6. Both food-related stimuli (burger and roast chicken) are meat. It is unknown how many respondents were on a vegetarian diet, which makes it impossible to assess the preferences or attractiveness of these stimuli. For a vegetarian, meat may arouse disgust rather than desire. How was this fact controlled in the study? Furthermore, both "low-calorie food stimuli" are medium rather than "low". Lettuce, cabbage or spinach would be a much better choice for low-calorie dishes than high-caloric and sweet fruits such as pear or grape.

7. It is unclear how the authors divided the sample into three groups, based on the Dutch Eating Behavior Questionnaire (DEBQ). The original study by Van Strien et al. (1986) developed a 33-item questionnaire with three factors (dimensions): (1) emotional eating (13 items); (2) external eating (10 items); and (3) restraint (10 items). No "inhibitory eating" is presented in any validation study in any language (Turkish, German, Italian, Polish, etc.). Therefore, the groups (successful restrictive dieter, failed restrictive dieter, and nonrestrictive dieter) are spilled on the wrong assumption and cannot be considered for further statistical analyses.

8. A separate chapter should be included with all statistical analyses explained, all variables listed in each model (including DV and IV), name of post-hoc tests (e.g. Tukey, Bonerroni, etc.), and criteria for effect size interpretation. 

Author Response

see annex

Reviewer 2 Report

Comments and Suggestions for Authors

The manuscript by Jiang et al titled “The absence of attentional bias to low-calorie food stimuli in restrictive dieters: Differences in the allocation of attentional resources to high-calorie foods” focused on the effects of high-/low-calorie foods on attentional bias in restrictive dieters.

I found the topic very interesting and curious, but there are several aspects that should be carefully reviewed by the authors.

1)      L 45-46: this concept is no different from the previous sentence. The two sentences could be combined into one and the text would be more streamlined

2)      L 57-58: “118ms/stimulus”. How is this time established?

3)      the authors should explain what lag2/lag4/lag7 are

4)      L 180-187: I suggest the authors move the paragraph to the M&M section, where statistical analyzes are more appropriate. In fact, these are not results

5)      The entire results section is impossible to follow without the help of figures that can quickly represent the discoveries. It is essential, for me, that the results are easily understandable without having to schematize what you read.

6)      Figure 2: what does this figure refer to? there is no reference in the text.

7)      what drives restrictive drivers who wants to lose weight and shape their body, as reported by the authors, to have a strong desire to eat?

8)      L 313: the hedonic contribution of food is linked to the olfactory sensitivity of individuals. Have the authors considered this aspect? Could difficulties in following a diet be linked to chemosensory aspects?

Author Response

see annex

Reviewer 3 Report

Comments and Suggestions for Authors

It is a very clear and scientifically sound study. However, it would be advisable to make certain improvements, which I describe below:

- It would be advisable to better define the sample in section 2.1, providing, for example, percentages of the three groups of diets followed by means of a bar chart or even incorporating a population pyramid.

- The text reports the internal consistency of the DEBQ, but does not specify which indicator is measured (Alpha, Omega...).

- Tables and figures should be presented in the text. For example: below Table X shows...

- In the Results section, I would recommend putting the data in tables. This makes it much easier for the reader to understand the results than with the results incorporated in the text.

- Figure 2 needs a clear legend for each coloured line, not just the acronym.

- Limitations to the study are presented in a strange way. It looks like a subsection of the Discussion, as it is presented under a heading "Possible limitations of this study", but if it is a subsection it should be presented within the overall structure with its correct subsection format (as, for example, the Method subsections).

- The conclusions are too brief. It would be advisable to expand them, as there would be much more to conclude from the results.

- The references should be updated. Many of them are very old. A few more references that are more up to date should be added to the list.

- The references should be revised, as the MDPI format is not well done.

Author Response

see annex

Round 2

Reviewer 1 Report

Comments and Suggestions for Authors

The authors responded satisfactorily to the reviewer's comments.

Reviewer 2 Report

Comments and Suggestions for Authors

No comment